# Phylogenetics and Mobilization of Genomic Traits of Cephalosporin-Resistant *Escherichia coli* Originated from Retail Meat

**DOI:** 10.3390/pathogens13080700

**Published:** 2024-08-19

**Authors:** Ewelina Iwan, Magdalena Zając, Arkadiusz Bomba, Małgorzata Olejnik, Magdalena Skarżyńska, Bernard Wasiński, Kinga Wieczorek, Katarzyna Tłuścik, Dariusz Wasyl

**Affiliations:** 1Department of Omics Analyses, National Veterinary Research Institute, 57 Partyzantow, 24-100 Pulawy, Polandmolejnik@umk.pl (M.O.); wasyl@piwet.pulawy.pl (D.W.); 2Department of Microbiology, National Veterinary Research Institute, 57 Partyzantow, 24-100 Pulawy, Polandwasinski@piwet.pulawy.pl (B.W.); 3Faculty of Biological and Veterinary Sciences, Department of Basic and Preclinical Sciences, Nicolaus Copernicus University in Torun, 11 Gagarina St., 87-100 Torun, Poland; 4Department of Food of Safety, National Veterinary Research Institute, 57 Partyzantow, 24-100 Pulawy, Poland; kinga.wieczorek@piwet.pulawy.pl

**Keywords:** food safety, *Escherichia coli*, genomics, cephalosporin-resistance, ARGs transmission

## Abstract

Contaminations with cephalosporin-resistant *Escherichia coli* across the food chain may pose a significant threat to public health because those antimicrobials are critically important in human medicine. The impact of the presented data is especially significant concerning Poland’s role as one of the leading food producers in the EU. This work aimed to characterize the genomic contents of cephalosporin-resistant *Escherichia coli* (n = 36) isolated from retail meat to expand the official AMR monitoring reported by EFSA. The ESBL mechanism was predominant (via *bla*_CTX-M-1_ and *bla*_SHV-12_), with the AmpC-type represented by the *bla*_CMY-2_ variant. The strains harbored multiple resistance genes, mainly conferring resistance to aminoglycosides, sulfonamides, trimethoprim, tetracyclines. In some isolates, virulence factors—including intimin (*eae*) and its receptor (*tir*) were detected, indicating significant pathogenic potential. Resistance genes showed a link with IncI1 and IncB/O/K/Z plasmids. Cephalosporinases were particularly linked to ISEc9/ISEc1 (*bla*_CTX-M-1_ and *bla*_CMY-2_). The association of virulence with mobile elements was less common—mostly with IncF plasmids. The analysis of *E. coli* isolated from retail meat indicates accumulation of ARGs and their association with various mobile genetic elements, thus increasing the potential for the transmission of resistance across the food chain.

## 1. Introduction

*Escherichia coli* (*E. coli*) has a great capacity to accumulate genetic traits, including those linked to antimicrobial resistance and pathogenicity [1]. The genome of *E. coli* varies in size and consists of approximately 3000 core genes and between 1000 and 2000 accessory genes, which often reflects the genomic composition of ecological niches from which the isolate originated [2]. It harbors various mobile genetic elements (MGEs), facilitating the horizontal gene transfer (HGT) of multiple traits between bacterial species [3]. MGEs play a critical role in bacterial evolution and often carry crucial accessory genes associated with bacteria’s ability to cause disease or endure selective pressure [4]. They consist of insertion sequences (IS), transposons (Tn), gene cassettes, integrons, bacteriophages, and plasmids [5].

Antimicrobial resistance (AMR), detected in *Enterobacterales,* is on the rise, with *Escherichia coli* resistant to third-generation cephalosporins being ranked second on the WHO Bacterial Priority Pathogens List, 2024 [6]. *E. coli* is commonly used as an indicator bacteria for the monitoring of the food production chain [6,7]. In the enterobacterial gene pool, *E. coli* may act both as a donor and a recipient of antimicrobial resistance genes—ARGs [2]. ARGs associated with MGEs can be easily transferred across the food production and distribution chain, creating a potential concern for consumers’ health [8]. Infections with multidrug-resistant bacteria have been associated with limited therapeutic options, prolonged duration of infection and a high rate of treatment failure [6]. In recent years, a significant spread of resistance genes has been seen, both due to environmental pressure associated with antimicrobial use and increased mobilization of resistance genes.

Resistance to β-lactams, including cephalosporins, is one of the primary mechanisms with an increasing trend among Gram-negative bacteria [5]. Cephalosporins are critical antimicrobials for the treatment of bacterial infections in both human and veterinary medicine [9,10,11]. Third- and fourth-generation cephalosporins, in particular, are critical antimicrobials for treating bacterial infections in both human and veterinary medicine. According to the WHO List of Medically Important Antimicrobials, they are currently classified as Highest Priority Critically Important Antimicrobials (HPCIA) [12]. The European Medicine Agency (EMA) categorizes these antibiotics into category B “Restrict” indicating their usage only for the treatment of clinical conditions, with no alternative antibiotics in Categories C or D [13]. The efficacy of cephalosporins is severely challenged by the increasing number of hydrolytic enzymes produced by bacteria to overcome an antibiotic effect [14,15]. Among the mechanisms responsible for cephalosporin resistance, extended-spectrum β-lactamases (ESBLs) and AmpC-type cephalosporinases are of the greatest importance [16]. ESBLs hydrolyze penicillins and later-generation cephalosporins (e.g., cefepime, cefetamet, ceftiofur, cefoperazone, and cefquinome) [12,13]. They are mostly plasmid-located and defined by the *bla*_CTX-M_-like, *bla*_SHV_-like, and *bla*_OXA_-like gene families [17]. AmpC-type mechanisms are not hampered by β-lactam-inhibitors and are coded among others by genes from *bla*_CMY-like_, *bla*_ACC-like_, *bla*_DHA-like_ families [16,17,18]. In the last decade, reports of cephalosporin-resistance in food production chains have become more frequent worldwide, as well as in Poland [9,19,20,21]. Cephalosporin resistance is monitored in food-producing animals and meat across the EU under the EC decision No. 2020/1729 (https://eur-lex.europa.eu, accessed on 1 March 2024). Cephalosporinase gene profiles found in bacteria isolated from humans and those across the food chain in Europe differ, thus cross-transmission cases seem to be rare [2,20,22]. Nonetheless, animal production continues to serve as a reservoir for ESBL/AmpC-producing bacteria [19,20,22].

Another aspect that should be considered in association with the resistance of *E. coli* originating from food-chain production is the presence of pathogenic traits. Pathogenicity may result from multiple combinations of diverse sets of virulence factors (VFs), mostly corresponding to colonization and toxicity during host infection [2]. Opportunistic infections also can be caused by atypical commensal *E. coli* strains, often with complex combinations of acceded VFs [1]. Pathogenic *E. coli* can commonly be classified into diarrheagenic and extraintestinal pathogens, with diverse pathotypes and various natural hybrid strains [2,23,24]. Moreover, additional risk due to the transfer of pathogenic traits to other bacterial species across environments should be considered [25].

Retail meat is a crucial link in animal food production, with a high risk of bacteria and their genetic load being transmitted to the consumers. Most of the foodborne outbreaks related to meat and meat products in Europe between 1989 and 2015 were caused by verotoxin-producing *Escherichia coli* (mostly O157H7) and *Salmonella* (mostly Typhimurium) [26]. The application of whole-genome sequencing, in addition to traditional monitoring, provides a broader perspective for the characterization of resistance. The impact of the presented data is especially important due to Poland’s role as one of the main EU food producers [11]. According to Eurostats in 2022, Poland was the main producer of poultry meat in the EU (21.0%), the fourth producer of pig meat (8.1%), and the fifth producer of beef (9.4%) (https://ec.europa.eu/eurostat/statistics-explained/index.php?oldid=4270960, accessed on 5 August 2024).

This study aimed to characterize the genomic content of cephalosporin-resistant *E. coli* found in Polish retail meat. The primary focus was on identifying resistance gene variants and virulence factors and understanding their potential spread and transmission through associations with mobile genetic elements (MGEs). This research seeks to expand upon the data reported in official antimicrobial resistance (AMR) monitoring reports submitted to the European Food Safety Authority (EFSA).

## 2. Materials and Methods

### 2.1. Collection and Selection of Samples

This study included a subset of 36 food isolates of *E. coli*, originating from beef (n = 5), pork (n = 12), and broiler meat (n = 19) and sequenced at the National Veterinary Research Institute (NVRI) for the ENGAGE project (ENA study ID: PRJEB23993). Since most of the strains were isolated from the official AMR monitoring in animals, the currently described isolates were obtained from a pilot study prior to the implementation of official AMR monitoring in foods (EFSA 2019). Fresh meat samples were collected at retail in the Lubelskie region of Poland between 2014 and 2016 and selectively screened (MacConkey agar supplemented with 2 mg/L of cefotaxime) for the presence of cephalosporin-resistant *E. coli* [20]. The confirmed ESBL/AmpC-positive *E. coli* (D68C Detection Set, Mast Diagnostic GmbH, Reinfeld, Germany) were whole-genome sequenced (WGS).

### 2.2. DNA Extraction

DNA was extracted using the Genomic Mini kit (A&A Biotechnology, Gdańsk, Poland) according to the manufacturer’s instructions. The quality and quantity of DNA were checked using a spectrophotometer (NanodropOne, Thermo Fisher Scientific, Waltham, MA, USA) and a fluorometer (HS and BR assay kit, Qubit 3.0, Thermo Fisher Scientific), respectively.

### 2.3. Library Preparation and Sequencing

Libraries were prepared from 1 ng of DNA according to Nextera XT kit (Illumina, San Diego, CA, USA) instructions. A dual indexing system (Illumina) was used for library labeling. Fragments of 400–800 bp were selected and cleaned up using AMPure magnetic beads (Beckman Coulter, Brea, CA, USA). The quality and quantity of each library were evaluated by both capillary gel electrophoresis (DNF-473 Standard Sensitivity NGS Fragment Analysis kit, Fragment Analyzer, Agilent, Santa Clara, CA, USA) and fluorimetry (BR assay kit, Qubit 3.0, Thermo Fisher Scientific). High-throughput sequencing was performed on the MiSeq (Illumina) platform with the use of a 2 × 300 bp V3 kit (Illumina).

### 2.4. Data Analysis

Data trimming was performed by Trimomatic 0.36 [27]. Reads were merged by BBMerge, then assembled by SPAdes software 3.13.0 [28,29]. The quality of genomic assemblies was checked with the QUAST tool [30]. Phylogenetic comparison of isolates was conducted based on the cgMLST matrix. An additional comparison was made with 42 sequences of *E. coli* from the meat production chain (beef carcasses) in Poland, available in the EnteroBase repository. Profiles of cgMLST and Allel matrix were generated by cgMLSTFinder-1.2 ver.: 1.0.1 (2021-08-29) according to Enterobase schema [31,32]. The phylogenetic trees (newick) were visualized using the iTOL online tool [33]. Sequence type was assessed using MLST ver.: 2.0.9 (2022-05-11, database: 2023-06-19) with the MLST allele sequence and profile data obtained from PubMLST [34]. Serotypes of *E. coli* were determined by Abricate 0.9.8 (Seemann T., https://github.com/tseemann/abricate, accessed on 1 March 2024) with the default version of the EcOH database [35]. Detection of specific markers was also performed by Abricate 0.9.8 with 80% identity and 80% minimum coverage thresholds using the CGE database: ResFinder: EFSA_2021 (2022-07-19), VirulenceFinder (2022-12-02), PlasmidFinder (2023-01-18). Point mutations were identified using the option: chromosomal point mutations—PointFinder (2022-08-08) of ResFinder 3.2, with default settings and the PointFinder database: EFSA_2021 (2022-04-22) [36]. The link between resistance genes, virulence factors and mobile genetic elements (plasmid replicons, transposons, and insertion sequences) was established by assessment of their proximity (up to 3000 bp flanking from each end of detected gene/gene cassette) on specific contigs via MobileElementFinder (MGE) v1.0.3 (2020-10-09) web tool with database ver.: v1.0.2 (2020-06-09) [37]. Selected genome fragments were visualized by SnapGene 5.0.7 and Proksee online tool [38]. Annotation was performed by Proksee: tool Prokka 1.14.6 and/or Bakta v1.8. [39,40]. Gene-MGE association was visualized by the online tool: SankeyMATIC (https://sankeymatic.com/build/, accessed on 10 March 2024).

## 3. Results

### 3.1. Molecular Typing

In total, twenty six sequence types were identified. The most commonly detected sequence types were ST354 (n = 4, broiler isolates) and ST10 (n = 3, single isolates from each source)—Appendix A. The most often predicted serotypes were O153:H34 (n = 3, in broiler), then O4:H27 (n = 2, in pork and beef), O32:H9 (n = 2, in pork and beef), and O51:H21 (n = 2, in broiler). In two cases, O antigen sequences were not detected—even after the detection thresholds had been lowered. All isolates showed a different number of cgMLST (the number of loci ranging from 2364 to 2408). The generated phylogenetic tree showed high diversity of *E. coli*, with two small clades distinguished in broiler strains, corresponding to ST354 (n = 4) and containing both ST1800 and ST7970 (n = 3) (Figure 1a). Similar diversity was identified in comparison with other Polish *E. coli* from the food-chain production included in this study (Figure 1b).

### 3.2. Acquired Resistance

The total number of resistance genes ranged from one (broiler meat isolate) to fourteen (found in pork) (Appendix A and Figure 1a). The majority of the isolates were multi-drug resistant (MDR), with the exception of two broiler *E. coli*. Most common antimicrobial genes coded resistance to the following: tetracyclines (*tetA/B*, n = 33), sulfonamides (*sul2*, n = 26), aminoglycosides (*aph*(6)-*Id*, n = 21 and *aph*(3″)-*Ib*, n = 21) and trimethoprim (*dfrA*, n = 23). Markers of plasmid-mediated resistance to fluoroquinolones (*qnrS1*/*qnrB19*), mostly in pork (n = 5) and less commonly in broiler (n = 2) and beef (n = 1) isolates were also detected. Additionally, one broiler *E. coli* carried the colistin-resistance gene *mcr-1.1*. In multiple cases (n = 16), the *bla*_TEM-1B_ gene, which encodes resistance to beta-lactams, was present in association with cephalosporinase-encoding determinants. In one case, *bla*_TEM-30_ was detected—an inhibitor-resistant beta-lactamase. The sequencing data confirmed the presence of various cephalosporin resistance genes across the studied group. The most common were genes coding the ESBL profile: *bla*_CTX-M-1_ (n = 15) and *bla*_SHV-12_ (n = 9). While *bla*_CTX-M-1_ was prevalent across all isolation sources, *bla*_SHV-12_ was present exclusively in poultry isolates. Less commonly, *E. coli* harbored other ESBL markers: bla_CTX-M-27_ (n = 2), *bla*_CTX-M-14_ (n = 1), *bla*_CTX-M-15_ (n = 1), *bla*_CTX-M-55_ (n = 1). AmpC phenotype was conditioned via *bla*_CMY-2_ (n = 7) across all the isolation sources. Sequences of 25 isolates harbored point mutations in the quinolone resistance-determining regions (QRDR) of *gyrA* (p.S83L and/or p.D87N, n = 25) and *parC* (p.S80I, n = 13). Six *parC* + *gryA* mutants also showed S458A mutation in *parE*—Appendix A. Most strains had at least two point mutations, indicating resistance to both quinolones, such as nalidixic acid, and fluoroquinolones (i.e., ciprofloxacin).

### 3.3. Virulence Factors

All isolates carried virulence factors—from 12 to 42 genes (Appendix A and Figure 2). The uppermost value (median) of detected VF was in the broiler isolates (n = 31) and the lowest in the *E. coli* from beef (n = 21). The most commonly noted were those associated with fimbriae formation: *yeh* fimbrial gene cluster (*yehC*, n = 36, *yehD*, n = 35, *yehB*, n = 34, *yehA*, n = 30), type 1 fimbriae—*fimH* (n = 34), curlin major subunit (*csgA*) n = 36, as well as long polar fimbriae (*lpfA*, n = 21). In addition, virulence genes codding tellurium ion resistance protein (*terC*, n = 36), lipoprotein NlpI precursor (*nlpI*) n = 36, increased serum survival (*iss*) n = 31, AraC negative regulator (*anr*) n = 30, outer membrane protease (*ompT*) n = 29, iron transport protein (*sitA*) n = 26, and the enterobactin siderophore receptor protein (*iroN*) n = 22 were frequently found across all isolates groups. Several genes codding bacteriocin: *cib* (n = 15), *cma* (n = 13), *cia* (n = 4), *cea* (n = 3) were present across the *E. coli* from all sources. Most isolates also harbored avian *E. coli* hemolysin (*hlyE*) n = 34 and hemolysin F (*hlyF*, n = 22) genes. Several VF were primarily presented in isolates from broiler meat: air—enteroaggregative immunoglobulin repeat protein (n = 5), *ibeA*—invasin of brain endothelial cells (n = 4), *astA*—heat-stable enterotoxin EAST-1 (n = 10, of which n = 2 in pork isolates), *eilA*—homolog of *Salmonella* main regulator of the pathogenicity island 1 (*HilA*) (n = 8, including single pork strain), and *espY2*—non-LEE-encoded type III secreted effector (n = 8). No genetic markers for Shiga-toxin were detected. Two isolates originated from pork harbored intimin genes (variant *eae*-e07-xi and variant *eae*-b01a-βξ) and its receptor (*tir*). In addition, these strains also carried other markers for adhesion and toxicity: *ehxA*—enterohaemolysin, *espA/B/F*-associated signal transduction, *nleA/B*—non-LEE encoded effectors, gad-glutamate decarboxylase, and the *iss* gene.

### 3.4. Plasmids

Overall, thirty-four types of plasmid replicons were identified. The highest median number of detected plasmid replicons per strain was in the *E. coli* isolated from broiler meat (n = 5) and the lowest from beef (n = 3)—Appendix A and Figure 3. The most commonly detected plasmid replicons across all isolation sources belonged to the IncF type, which consisted mostly of IncFIB (n = 32), IncFII (n = 12), and IncFIC (n = 9). Commonly detected were also IncI1 (n = 21), IncB/O/K/Z (n = 11), IncX1 (n = 8), and IncX3 (n = 6) replicons. The sequencing data also indicated the presence of Col-type plasmids: MG828 (n = 17), RNAI (n = 17), and pHAD28 (n = 10), with MG828 present in most broiler isolates.

### 3.5. MGE Associated with Resistance and Virulence Genes

In 30 analyzed strains, at least one AMR gene was linked to an MGE (a plasmid replicon or IS/Tn). Resistance genes were primarily associated with the plasmid replicons IncI1 (n = 10), IncB/O/K/Z (n = 7), and IncN (n = 6), and less often with IncQ, IncN, IncFII, IncFIB, IncX1, and IncX4 (Figure 4a,b). Some of the strains also harbored multi-resistance (MDR) cassettes in the proximity of multiple MGEs: insertion sequences, transposons, and plasmid replicons, e.g., IncN (Figure 5a) and IncB/O/K/Z (Figure 5b). In the single presumably colistin-resistant strain, *mcr-1.1* was located on the same contig as the IncX4 replicon; the BLAST results showed a 100% identity match with 99.9% coverage to the reference plasmid ac. No. MK869757.1 (*Escherichia coli* strain MFDS2258 plasmid pMFDS2258.1) (Figure 5c). ARGs were also linked to multiple insertion sequences: ISVsa3, IS102, ISSbo1, ISEc9, ISKpn19, Tn2, ISKpn19, IS26, ISEc32, IS629. Close associations were observed between ISVsa3 and *floR* (n = 5) and between ISKpn19 (n = 5) and *qnrS1* (direct proximity up to 1276 bp) (Figure 4a).

The markers of cephalosporin-resistance were mostly linked to IncI1, via *bla*_CTX-M-1_ (n = 5); IncB/O/K/Z, via *bla*_CMY-2_ (n = 3); but also with IncX1, via *bla*_SHV-12_ (n = 1); and IncX1, via *bla*_CTX-M-1_ (n = 1) (Figure 4b). In 27 out of 36 strains, insertion sequences were detected in direct proximity to the cephasporinase genes. Most genes encoding resistance to cephalosporins—via *bla*_CTX-M-1_, *bla*_CTX-M-15_, *bla*_CMY-2_, and *bla*_CTX-M-14_—were in direct proximity to insertion sequences, particularly with ISEc9 (n = 20). In addition, there were cases of association (proximity up to 3 kbp) of *bla*_CTX-M-1_ with ISVsa3 (n = 1), *blaCMY*_-2_ with ISSbo1 (n = 1), and *bla*_CTX-M-27_ with IS102 (n = 2) (Figure 4b).

In addition, in most of the analyzed strains (n = 33), at least one VF was linked to some kind of MGE (Figure 6). With some exceptions (e.g., *anr* with Tn6082), most of the virulence genes were not linked to IS/transposons. In seven separate cases, THE plasmid replicon was located together with both resistance and virulence genes on the same contigs. Virulence factors *traT*, *ompT*, *etsC*, *hlyF*, *anr*, *traJ* showed a strong link with the IncF plasmid group (n = 42), mostly FIB (AP001918) (n = 22), FII (n = 12), and FIC(FII) (n = 6). In one case, a single contig of the *E. coli* genome harbored multiple virulence factors (*iroN*, *mchF*, *etsC*, *ompT*, *etsC*, *cvaC*, *iss*, *anr*, *tsh*, *hlyF*) with IS and two plasmid replicons: IncFIB and IncFIC, possibly indicating the presence of a. multi-replication plasmid (Figure 7).

## 4. Discussion

In 2019, the EFSA recognized the importance of WGS (whole-genome sequencing) for identifying and monitoring emerging health threats from food-borne pathogens, particularly for outbreak investigation, source attribution, and risk assessment (EFSA 2019). The role of geographically diverse genomic data provided by various stakeholders is crucial for the comprehensive utilization of genomic methods in food safety. This study focuses on the genomic data from a limited number of cephalosporin-resistant *E. coli* collected from retail meats in Poland in 2014–2016. This period predates the more common use of high-throughput sequencing (HTS) technology in antimicrobial resistance monitoring in Poland. The in-depth analysis offers valuable insights for risk assessment and helps describe the spread of genomic traits across the food production chain, despite the strains being from previous AMR monitoring. The presented data is especially valuable, given Poland’s significant role as a food producer in the EU—particularly in the context of poultry production (https://ec.europa.eu/eurostat/statistics-explained/index.php?oldid=4270960, accessed on 5 August 2024) [11].

Most isolates were of various sequence types, showing no association with any specific *E. coli* serotype or source of isolation, indicating significant genomic variability. However, a small clad of AmpC and ESBL carriers of ST354 *E. coli* from broiler meat was observed. This clonal lineage has been associated with poultry and broiler meat, indicating that farm animals may be reservoirs of cephalosporin-resistant and pathogenic *E. coli* [41,42]. Some of the identified genotypes—ST10 and ST648—are predominantly associated with ESBL *E. coli*, but in our study, they were found to be both ESBL (ST10) and AmpC (ST648) producers [43]. The Core Genome Multi Locus Sequence Typing (cgMLST) analysis showed further variability in the genomic composition of the collected isolates. This is consistent with the data of other *E. coli* originating from food-producing animals in Poland (EnteroBase).

Genomic bases of cephalosporin resistance were confirmed in all surveyed *E. coli*. The ESBL mechanism was present in isolates from all types of meats and prevailed over AmpC, which was present only in pork and broiler meat. Similar results were shown by other studies regarding food animals in Poland (broilers, layers, turkey, pig, and cattle), where cephalosporin resistance mechanisms were detected in multiple species, except cattle [19]. ESBL prevalence over AmpC is typical for multiple EU/EEA countries [44]. The main gene families associated with resistance to cephalosporins in characterized isolates were CTX-M-, SHV- and CMY-encoding gene groups, with the ESBL variant—*bla*_CTX-M-1_—predominating in pork and beef isolates. This is consistent with recent data from the EFSA report, where, in general, the *bla*_CTX-M-1_ variant was a dominant marker responsible for cephalosporin resistance in ESBL-/AmpC-producing *E. coli* from animals and meat from Italy, Germany, Czechia, and Finland [9]. The strong link between this variant and both livestock and food of animal origin was confirmed by scientific sources across Europe [45].

The second most predominant gene variant responsible for resistance to cephalosporins (ESBL) was *bla*_SHV-12_, detected only in broiler meat isolates. This seems to be in accordance with other regional and European data (Poland, Germany, Spain, and Denmark), where *bla*_CTX-M-1_ and *bla*_SHV-12_ are dominant in livestock- and poultry-based food production chains, respectively [16,19,20,22,45]. In *E. coli* from beef and pork (*E. coli* ST10), we also detected the presence of *bla*_CTX-M-27_, the prevalence of which is on the rise [9,46]. The *bla*_CTX-M-27_ gene has arisen as a single-nucleotide mutation of *bla*_CTX-M-14_ and is now emerging globally [47]. Cases of *bla*_CTX-M-27_-positive strains originating from animals, food-chain production, and humans were described worldwide, showing its global spread [48,49]. We also noted the single case of blaCTX-M-55, which is the second most common ESBL-encoding gene in the *E. coli* of human and animal origin in Asia, recently also reported in *E. coli* from cattle in Europe—Germany and Italy [9,50]. The CTX-M-14 and CTX-M-15 variants of the *bla* gene are confirmed to be the most globally spread, mostly associated with cases of infections in humans, but also, in the case of *bla*_CTXM-15_, with cattle [9,45,46]. However, in our data, only single cases of those variants were detected in pork and beef samples. This seems to confirm the hypothesis that ESBL/AmpC profiles occurring in the public health sector are significantly different from those of animal husbandry and food production, with occasional cases of cross-sectoral spillage [51].

The AmpC mechanism was present only in *E. coli* originating from broiler meat and pork and coded exclusively via *bla*_CMY-2_. This gene variant is the most commonly reported plasmid-mediated AmpC mechanism in European food-producing animals—mainly in poultry [9]. Some studies have shown that *E. coli* isolated from poultry and human infections share a matching cephalosporin resistance profile, suggesting that the transmission of resistance from poultry meat to humans via the food chain may occur [52].

MGEs represent some of the challenges in counteracting the dissemination of antimicrobial resistance. They contribute to the spread of relevant resistance determinants and promote horizontal gene transfer among bacteria [53]. In our study, on multiple occurrences, the detected cephalosporinases were mobilized via various MGEs, with some regularities.

ARG mobilization via plasmid is underestimated due to the limitation of short-read sequencing. However, in multiple cases, we were able to detect a link between resistance markers and plasmids. In our collection, the *bla*_CTX-M-1_ gene was strongly associated with the replicon of the IncI1 plasmid, which is frequently described as a vector harboring and mobilizing ESBL mechanisms compartments in the food production chain [21,22,45,54,55,56]. Other studies indicate the association of the ESBL marker—*bla*_SHV-12_—with the IncX1 plasmid but also plasmids of other groups: IncI1, IncK, IncF, IncX3. We found a single case where bla_SHV-12_ was located on IncX1 [5,16,46,57,58,59,60,61]. To our knowledge, this is the first case of an IncX1 plasmid harboring *bla*_SHV-12_. Most often, IncX1 plasmids are linked to the *bla*_TEM_, *bla*_CMY-2_, and QNR genes [61,62,63,64]. The AmpC mechanism—*bla*_CMY-2_ originates from its chromosomal variant and has since been associated with different plasmids—IncK, IncI1, IncA/C, and IncFIA-FIB [5,18,56,65]. In a few cases described in our study, *bla*_CMY-2_ was linked to the IncB/K/O/Z plasmids (*E. coli* from broiler meat and pork). Recent retrospective comparisons of genomic data showed a similar link of the *bla*_CMY-2_ variant to IncB/K/O/Z plasmids harbored by *E. coli* isolated in 2010 from poultry production in Germany [66]. While many ESBL/AmpC markers were not linked to plasmids, the strain compositions showed the presence of multiple replicons: IncI1, IncF, IncFIB, IncFII, IncFIC, IncB/O/K/Z, IncX1, IncX3, indicating the possible existence further associations with plasmids.

We determined that the variants *bla*_CTX-M-1_, *bla*_CTX-M-14_, *bla*_CTX-M-15_, and *bla*_CMY-2_ of cephalosporinases were strongly associated with ISEc9 (ISEc1). This mobile element has been identified not only as responsible for disseminating various ESBL/AmpC genomic markers but also as a strong promoter for the expression of *bla*_CTX-M_, advancing the evolutionary success of its host strains [46,60]. ISEcp1 has also been reported to be able to pick up regions of different lengths in independent transposition events. Thus, it can simultaneously move adjacent pieces of DNA of different origins. It appears to have been responsible for capturing many different resistance genes in numerous cases from known source organisms [5]. Interestingly, detected cases of *bla*_CTX-M-27_ were mobilized via IS102. This insertion sequence was also reported in direct proximity of *bla*_CTX-M-65_ in cephalosporin-resistant *E. coli* and *Salmonella enterica* strains of animal origin [37,67]. To the best of our knowledge, this is the first report of *bla*_CTX-M-27_ mobilization via IS102. Increased mobilization via multiple plasmids and IS (especially Ecp9/Ecp1) seems to be another key element in the worldwide spread of *bla*_CTX-M_/*bla*_CMY_ [22,47,56,60,67]. We also noted the occurrence of *bla*_SHV-12_ mobilization via IS26, which seems to be in agreement with the literature data [5,60]. In addition, some *bla*_SHV-12_-positive isolates harbored multi-resistance gene cassettes mobilized via plasmids (IncFII and/or IncI1) or insertion elements (IS26). The multidrug-resistant genotype was frequently linked to *bla*_SHV-12_-positive *E. coli* from poultry and the associated food-chain production [58,59,65].

Characterized *E. coli* harbored a broad range of ARGs, with narrow-spectrum beta-lactamases (*bla*_TEM-1B)_ being the most common. Most of the surveyed strains were classified as multidrug-resistant (MDR). Isolates often harbored combinations of co-resistance to aminoglycosides, fluoroquinolones, phenicols, sulfonamides, tetracyclines, and trimethoprim. Genes coding aminoglycoside resistance were frequently co-located with each other (*aph(6)-Id* with *aph(3″)-Ib* or *aph(3″)-IIa* with *aph(4)-Ia*) or/and various AMR (e.g., sulfonamide, tetracycline, or trimethoprim), creating large multi-resistance cassettes. According to the global trends, MDR in food animals is rising, although the recent EU data seem more optimistic [44,68]. In the literature, MDR occurrence could be attributed to co-resistance to antimicrobials commonly used in agriculture, such as tetracyclines, streptomycin, and sulfonamides, which were associated with high resistance rates [2,6,9,69].

Multiple ARGs were frequently mobilized via IS, transposons, and/or plasmid replicons (often multiple replicons), creating variable regions rich in mobile genetic elements, ARGs, and often metal resistance. In several cases, the potential presence of a multi-replicon plasmid was detected. Those additional replicons may increase the fitness cost of harboring the plasmid, but they may also be advantageous, e.g., by increasing the range of bacterial host species [5]. We noted a case of *E. coli* from broiler meat carrying resistance determinants to both cephalosporins (*bla*_CTX-M-1_) and polymyxin (*mcr-1.1*). Furthermore, the *mcr-1* gene was linked to plasmid IncX4, suggesting its mobilization. Recently, reports of plasmid-mediated movable colistin resistance have become frequent worldwide [70,71]. In Poland, Zając et al. (2019) reported *mcr-1* associated with IncX4 and IncHI2 in turkeys, chickens, pigs, and cattle, displaying the scale of the problem [72]. The presence of colistin ARG in the same bacterial species from different compartments, including food-producing animals, food, humans, and the environment, indicates a probable transmission between ecological niches [6,73]. Plasmid-mediated fluoroquinolone resistance (*qnrS1*/*qnrB19*) was the most common in pork isolates, which differs from the previous findings of Wasyl et al. (2013) concerning food animals from this region. Those authors noted the highest percentage of resistant strains in poultry (broilers, layers, and turkeys) while the lowest percentage was in cattle [20].

The data draw attention to the co-occurrence of diverse genomic traits—MDR, VF, and MGEs. Genes associated with pathogenicity may encode, among others, activities such as adhesion, invasion, attachment, iron acquisition, motility, and toxin activity [23,24]. Across all isolation sources, we observed complex, mosaic patterns of virulence genes, with markers associated with some of the pathogenic strains, which were not classifiable as one particular pathotype. This observation seems to be in accordance with the more recent view on the classification of *E. coli* pathotypes [24]. Some of the isolates in this study (mostly from pork and broiler meat) harbored *astA*, which encodes the enteroaggregative *E. coli* heat-stable enterotoxin (EAST1), repeatedly reported in pathotypes: enteroaggregative (EAEC), enterotoxigenic (ETEC), and enteropathogenic *E. coli* (EPEC) of human or animal origin [74]. On average, the beef strains contained fewer virulence factors; they mostly carried genes linked to adhesion and very few genes were associated with toxicity. The broiler isolates harbored a significant number of genes coding bacteriocins (colicin), the enteroaggregative immunoglobulin protein (*air*), the *Salmonella* HilA homolog (*eilA*), and the EAST1 toxin. Those genes have been reported to enhance the fitness of bacteria in a wide range of habitats and to be involved in adhesion and intestinal colonization [75,76,77]. Interestingly, two pork *E. coli* strains with the highest number of virulence factors coded multiple genes from the pathogenicity island LEE (locus of enterocyte effacement), associated with STEC O157:H7. This chromosomal pathogenicity island (35 kb) harbors a type III secretion system (TTSS), the adhesion molecule intimin, and its receptor, which are linked to highly pathogenic strains of the Shiga toxin (*stx*)-producing *Escherichia coli* (STEC) [78]. Both isolates were negative for *stx* but encoded intimin (*eae*), the translocated intimin receptor (*tir*), the type III secretion system (*espF*, *espA*), and secreted protein B (*espB*). This combination of LEE markers enables AE (attaching and effacing) lesions and is often present in enterohemorrhagic (EHEC) and enteropathogenic *E. coli* (EPEC) [78,79]. In addition, both isolates harbored the non-LEE effector A-C (*nleA*, *nleB*, *nleC*), which are associated with HUS (hemolytic uremic syndrome)-causing strains of EHEC and VF associated with plasmids—enterohemolysin (*ehxA*, *espP*), catalase peroxidase (*katP*) [80]. Overall, those pork isolates had significant pathogenic potential, which, combined with a multi-resistant profile, creates serious health risks in case of transition and subsequent infection.

In addition, we observed cases where pathogenicity traits (*anr, traaT, estsC, hlyF, traJ*) were linked to plasmids, mostly from the IncF group (FI/FII/FIC/FIB). Epidemic IncF plasmids are the most abundant plasmid types among *E. coli*, especially within the MDR extraintestinal pathogenic *E. coli* (ExPEC), such as ST131 and ST410 [81]. IncF plasmids commonly harbor addiction systems, promoting their stability and maintenance in the bacterial host in different environmental conditions [53]. In addition, some virulence traits of cephalosporin-resistant *E. coli* were shown to be linked to the insertion sequence, especially 629 (IS3 family member). IS629 is reported to be one of the reasons for the evolutionary success of EHEC O157:H7 and the enterotoxigenic *E. coli* (ETEC) O139 and O149 serotypes. According to the literature, IS629 promotes genome plasticity and genetic diversity among the STEC strains, enhancing their abilities to adapt to hostile environments and rapidly take up virulence factors [82].

## 5. Conclusions

The *E. coli* isolated from retail meat showed significant genomic diversity and complex genomic compositions. ESBL (encoded mostly by *bla*_CTMX-M-1_ and *bla*_SHV-12_) was identified as the predominant phenotype responsible for resistance to cephalosporines, with AmpC (via *bla*_CMY-2_) in the minority. The genomic data indicate the strong association of ARGs with MGEs, thus increasing the potential for transmission of these traits across the food chain. Further, some of the isolates had a significant pathogenic potential, which, combined with a multi-resistant profile, creates serious risk in case of transition and subsequent infection. This research underscores the complexity of antimicrobial resistance dynamics, emphasizing the importance of comprehensive genomic surveillance and understanding the role of mobile genetic elements in resistance dissemination.

## Figures and Tables

**Figure 1 pathogens-13-00700-f001:**
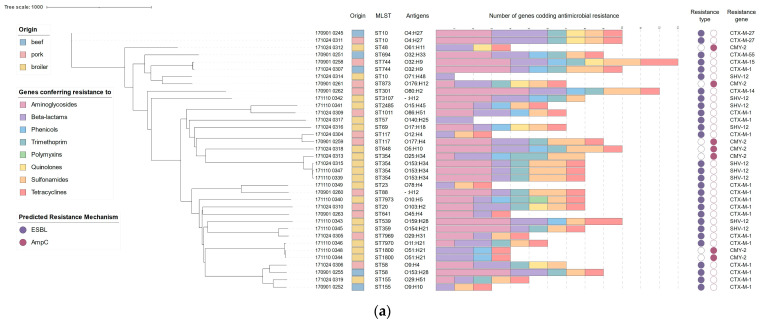
(**a**) Phylogeny of characterized *E. coli* from retail meat, based on core genome markers (cgMLST). The presented dataset describes the origin, sequence type, identified antigens, number of detected ARGs from each antibiotic class, and mechanisms of resistance to cephalosporins associated with identified gene variants. (**b**) Core genome (cgMLST) phylogenetic tree of *E. coli* strains characterized across this study (n = 36), color-coded in accordance with isolation source, as well as *E. coli* sequences (n = 42) from Polish meat production chain (beef carcasses), acquired from EnteroBase (color-coded green).

**Figure 2 pathogens-13-00700-f002:**
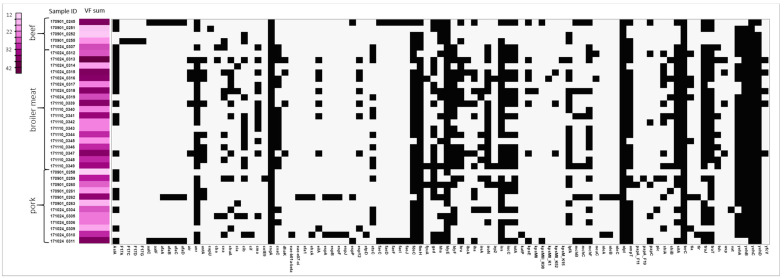
The matrix summarizes the presence of virulence factors (VFs) in the genomes of *E. coli*, categorized by their isolation sources: beef, broiler meat, and pork. In the matrix, the presence of each VF is indicated in black. The distribution of VFs across strains is represented by shades of violet, with the total number of detected VFs per strain ranging from a minimum of 12 (lilac) to a maximum of 42 (purple), as indicated in the legend.

**Figure 3 pathogens-13-00700-f003:**
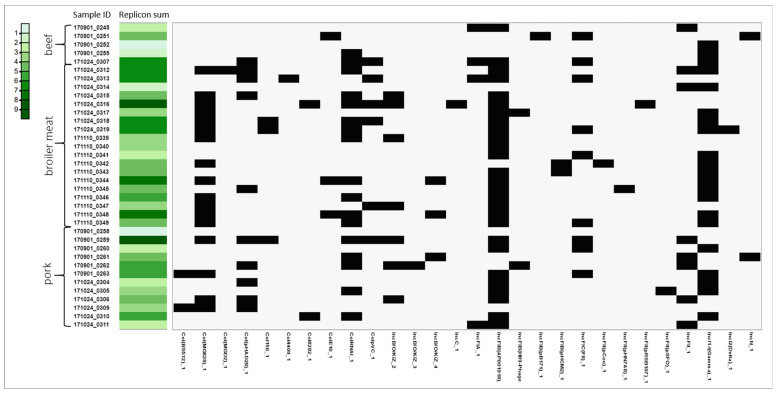
The matrix summarizing the presence of plasmid replicons in *E. coli* genomes grouped according to isolation sources—beef, broiler meat, and pork. The presence of replicons is indicated in black on the matrix. The distribution of green color represents the total number of detected plasmid replicons per strain, ranging from light green (minimum, n = 1) to dark green (maximum, n = 9), as illustrated in the legend.

**Figure 4 pathogens-13-00700-f004:**
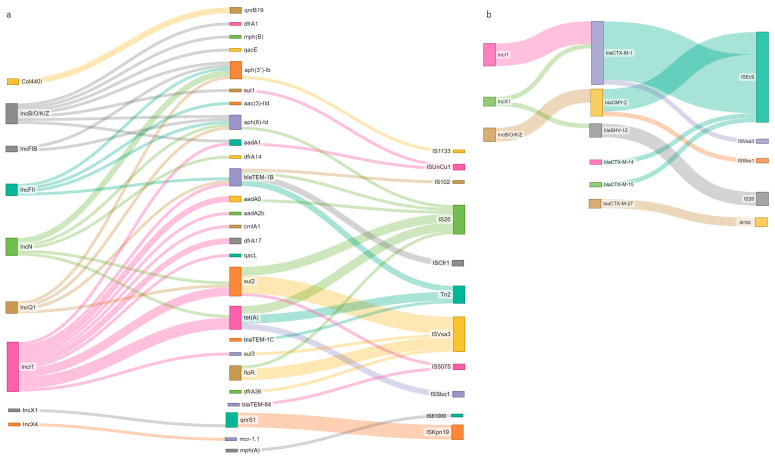
(**a**,**b**) Association between antimicrobial resistance genes and MGEs, both plasmid replicons and IS/Tn; (**a**) ARGs of all antibiotic classes, with the exception of cephalosporins, which are presented in (**b**). The link between the resistance gene and MGEs was established when both were in direct proximity (up to 3 kb) in genome assembly.

**Figure 5 pathogens-13-00700-f005:**
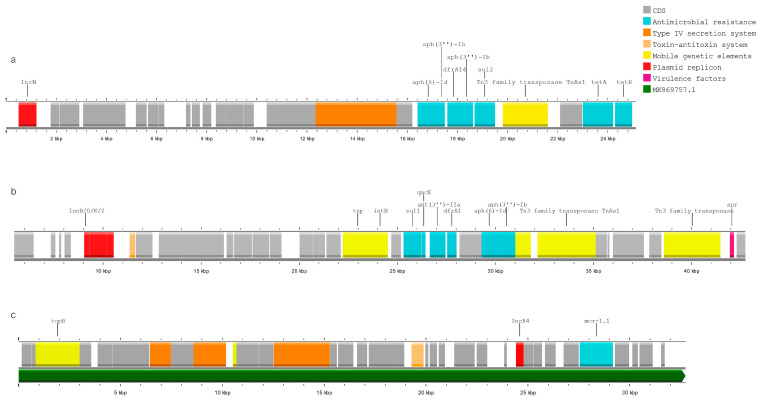
(**a**–**c**) Annotated fragments of *E. coli* genomes that contain resistance genes linked to MGEs. (**a**) Sample 170901_0261, harboring IncN replicon with multiple ARGs (*aph*(6)-Id, *aph*(3″)-Ib, *dfrA1*, *aph*(3″)-Ib, *sul2*, *tet*A/R); (**b**) sample 170901_0259 harboring IncB/O/K/Z replicon with MDR cassette (*dfrA1*, *aadA1*, *aph*(6)-Id, *aph*(3″)-Ib, *qacE*, *sul1*); (**c**) sample 171110_0340 harboring IncX1 replicon with *mcr-1.1* gene, aligned with reference sequence of mcr-plasmid MK869757.1 (color-coded green).

**Figure 6 pathogens-13-00700-f006:**
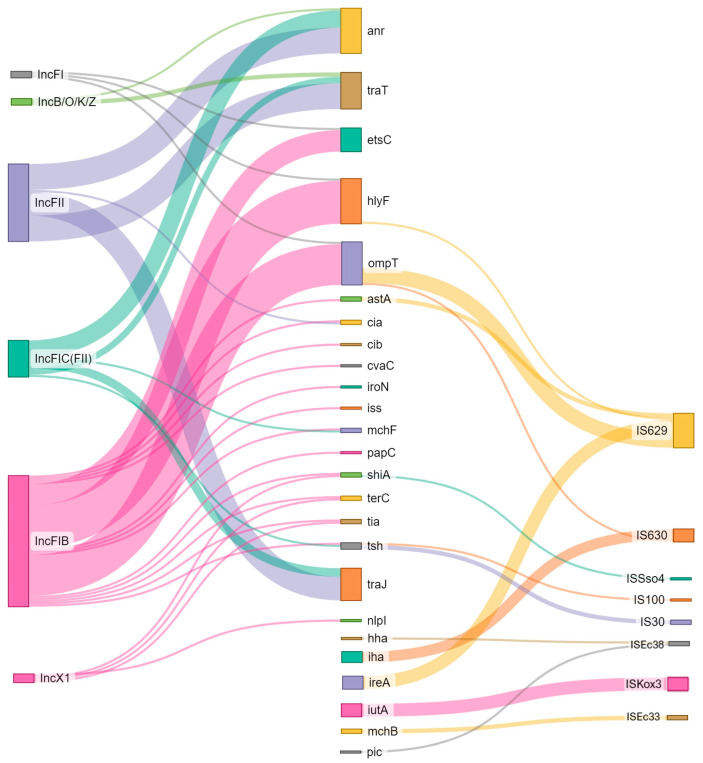
Association between virulence factors and MGEs, both plasmid replicons and IS/tn. The link between VF and MGE was established when both were in direct proximity (up to 3 kb) in the genome assembly.

**Figure 7 pathogens-13-00700-f007:**
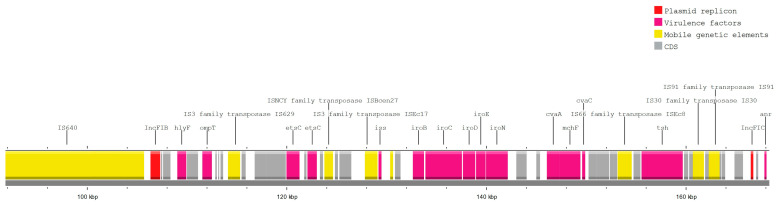
Annotated fragment of the strain 171110_0341 genome, harboring virulence factors *(iroN*, *mchF*, *etsC*, *ompT*, *etsC*, *cvaC*, *iss*, *anr*, *tsh*, *hlyF*) with multiple IS and two plasmid replicons: IncFIB and IncFIC.

## Data Availability

The DNA sequences (fastq) from the isolates were deposited in the Nucleotide Archive (ENA) under project numbers: ERS2055342, ERS2055348, ERS2055349, ERS2055352, ERS2055355, ERS2055356, ERS2055357, ERS2055358, ERS2055359, ERS2055360. ERS2055362, ERS2055363, ERS2055364, ERS2055365, ERS2055366, ERS2055367, ERS2055368, ERS2055369, ERS2055370, ERS2055371, ERS2055372, ERS2055373, ERS2055374, ERS2055375, ERS2055376, ERS2055377, ERS2055378, ERS2055379, ERS2055380, ERS2055381, ERS2055382, ERS2055383, ERS2055384, ERS2055385, ERS2055386, ERS2055387.

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
