# Peer review of "Phylogenetics and Mobilization of Genomic Traits of Cephalosporin-Resistant Escherichia coli Originated from Retail Meat"

_pathogens, 2024, doi:10.3390/pathogens13080700_

Round 1

Reviewer 1 Report

Comments and Suggestions for Authors

The subject of the research has a great practical value, the article presents data important for public health. And AMR is the problem widely discussed in EU. The study was conducted on retail meat samples, from which 36 cephalosporin - resistant E.coli isolates were selected.

Abstract briefly presents the clues of the research.

lines 69-70 & 74-75 & 317-318 &351-352 - especially, but please check the whole manuscript and please put references in order

line 87 "last link" is disputable in this case, when we will look at the whole food chain

line 91 please confirm the role of Poland by statistical data

line 92 - any references form EU?

Part of Introduction dedicated to the chosen research material is too general and brief - should be extended.

Authors presented the aim of the study and related it to the EFSA monitoring report.

line 100 M&M part starts from isolates, we do not get detailed  information about the retail meat samples, which should be included. Collection of meat samples takes place in 2014-2016, so the analysis is not presenting the current situation.

M&M part is very brief and general. Data analysis methodology correct.

Figures legends are very general. Figures are correlated with text, and all represents important data. The quality of the figures is fine.

The Results part is detailed. I have no major comments to it.

The Discussion part is the strongest part of this manuscript, broadly analyzed own data in comparison with literature and EFSA reports. It shows the expertise and good knowledge of the Authors in the field of research, as well as the current state of art in EU.

Please underline the novelty & limitations of the study.

Conclusions are too general again.

DNA sequences were placed in repository.

The references list is broad and very actual, which also manifests the importance of the subject.

The research gave valuable data, and as Authors declared was the basis for further, ongoing project.

Author Response

The authors kindly thank the reviewer for the time and effort spent revising the manuscript. All suggestions were considered while preparing the corrected version of the publication, and all changes have been added to the revised manuscript in tracking mode. Additionally, further language proofing of the article was performed. In the attachment, the authors have responded to each of the reviewer's comments.

Thank You again for the valuable suggestions. The authors greatly appreciate it.

Kind regards,

Reviewer 2 Report

Comments and Suggestions for Authors

The manuscript titled "Phylogenetics and Mobilization of Genomic Traits of Cephalosporin-Resistant Escherichia coli Originated from Retail Meat" makes a significant contribution to our understanding of the potential spread and transmission of antimicrobial resistance genes and virulence factors through mobile genetic elements in E. coli from retail meat.

The findings of this manuscript are particularly noteworthy due to their potential implications for public health and food safety. The data presented illuminate the complex mechanisms underlying the spread of antimicrobial resistance, highlighting the importance of monitoring and addressing such occurrences within the food chain.

Furthermore, the manuscript aligns well with the journal's scope, as it explores the molecular intricacies of antimicrobial resistance in a context highly relevant to current global concerns.

However, the manuscript exhibits some deficiencies in several areas:

  1. Proper bacterial nomenclature, where italics are lacking.
  2. English writing.
  3. An excessively extensive bibliography (78)
  4. The results need to be structured and systematized. Additionally, including one or two tables summarizing the results would aid readers in comprehending the manuscript, as the data is exhaustively described in the text.
  5. The study designed ethics statements

Another few points could be improved, as follows:

Line 16: “due to those antimicrobials being critically important in medicine.” , it is suggested “………in human medicine”.

Line 33: “those linked with resistance and pathogenicity”,  it is suggested “….antimicrobial resistance ……….”

Line 43 and 44: “ Antimicrobial resistance (AMR) detected in E. coli is on the rise, the species is ranked third on the list of antibiotic-resistant “priority pathogens”. Here should be used the original reference from WHOwhere carbapenem-resistant Enterobacterales (CRE) and third-generation cephalosporin-resistant Enterobacterales (3GCRE) received the highest scores, solidifying their classification in the critical priority category of BPPL-2024.

(WHO Bacterial Priority Pathogens List, 2024: bacterial pathogens of public health importance to guide research, development and strategies to prevent and control antimicrobial resistance. Geneva: World Health Organization; 2024. Licence: CC BY-NC-SA 3.0 IGO.)

Line 57 and 58: “Cephalosporins in particular, are critical antimicrobials for treatment of bacterial infections in both human and veterinary medicine”, it is recommended to specify the third and fourth generations of cephalosporins that are critical to human medicine and clarify their legal use in veterinary medicine(Regulation EU 2019/6).

Line 72-76: “Cephalosporinase gene profiles that occur in bacteria isolated from humans and across the food chain in Europe differ; crosstransition cases occur but seem to be rare (Poirel et al. 2018; Ceccarelli et al. 2019; Wasyl et al. 2013). Nevertheless, animal production remains a reservoir for ESBL/AmpC-producing bacteria (Ceccarelli et al. 2019; Lalak et al. 2016; Wasyl et al. 2013).” - This paragraph should be improved.

Line 96: “genetic elements (MGEs) to expand the data form official AMR monitoring reported to”, it is suggested “…..from….”

Line 313: Reference: Meunier et al. 2006 - current results should not be compared with such old references.

Line 356: “To our knowledge this is the first case of IncX1 plasmid harbouring blaSHV-12. “the following article should be consulted and the statement corrected -  Aldea I, Gibello A, Hernández M, Leekitcharoenphon P, Bortolaia V, Moreno MA. Clonal and plasmid-mediated flow of ESBL/AmpC genes in Escherichia coli in a commercial laying hen farm. Vet Microbiol. 2022 Jul;270:109453. doi: 10.1016/j.vetmic.2022.109453. Epub 2022 May 16. PMID: 35640410.

Line 380-381: “. To the best of our knowledge, this is the first report of blaCTX-M-27 mobilization via IS102.”-  the following article should be consulted and the statement corrected -  Salinas L, Cárdenas P, Graham JP, Trueba G. 2024.IS26 drives the dissemination of bla CTX-M genes in an Ecuadorian community. Microbiol Spectr12:e02504-23.

https://doi.org/10.1128/spectrum.02504-23

Best Regards,

Author Response

The authors kindly thank the reviewer for the time and effort spent revising the manuscript. All suggestions were considered while preparing the corrected version of the publication, and all changes have been added to the revised manuscript in tracking mode. Additionally, further language proofing of the article was performed. In the attachment, the authors have responded to each of the reviewer's comments.

Thank You for the time and effort invested into improving this work. The authors greatly appreciate it.

Kind regards,

Reviewer 3 Report

Comments and Suggestions for Authors

The manuscript provides valuable insights into the genomic traits and antimicrobial resistance mechanisms of cephalosporin-resistant Escherichia coli isolated from retail meat in Poland. While the study's findings are significant and relevant to public health, the manuscript requires substantial revisions to improve grammar, clarity, and formatting according to the journal's guidelines.

-          Proofreading and Grammar: A thorough proofreading is essential to correct numerous grammatical errors and awkward phrasings throughout the manuscript. Consider enlisting the help of a professional editor or a colleague with strong writing skills.

-          Formatting: Ensure that the manuscript adheres to the journal's formatting guidelines. Italicize all scientific names (e.g., E. coli should be italicized throughout the manuscript). Revise all reference numbers to be enclosed in brackets and ensure they are correctly cited in the text.

Comments on the Quality of English Language

Extensive editing of English language required

Author Response

(The authors gave the same response as above.)

Round 2

Reviewer 2 Report

Comments and Suggestions for Authors

The revisions have significantly enhanced the manuscript, and all suggestions have been considered. the manuscript is suitable for publication in present form.

Reviewer 3 Report

Comments and Suggestions for Authors

The revised manuscript demonstrates improved writing, clarity, and formatting. The edits in the discussion section effectively convey the significance of this study, highlighting the importance of geographically diverse genomic data. I believe this data will be valuable not only for a single country but also for the EU and neighboring regions.